

# Constraining the particle-scale diversity of black carbon light absorption using a unified framework

Payton Beeler and Rajan K. Chakrabarty

Center for Aerosol Science and Engineering, Department of Energy, Environmental and Chemical Engineering, Washington University in St. Louis, St. Louis, MO 63130

Correspondence: Rajan Chakrabarty (chakrabarty@wustl.edu) and Payton Beeler (beelerpayton@wustl.edu)

**Abstract.** Atmospheric black carbon (BC), the strongest absorber of visible solar radiation in the atmosphere, manifests across a wide spectrum of morphologies and compositional heterogeneity. Phenomenologically, the distribution of BC among diverse particles of varied composition gives rise to enhancement of its light absorption capabilities by over twofold in comparison to that of nascent or unmixed homogeneous BC. This situation has challenged the modeling community to consider the full complexity and diversity of BC on a per-particle basis for accurate estimation of its light absorption. The conventionally adopted core-shell approximation, although computationally inexpensive, is inadequate in not only estimating but also capturing absorption trends for ambient BC. Here we develop a unified framework that encompasses the complex diversity in BC morphology and composition using a single metric, the phase shift parameter ($\rho_{BC}$), which quantifies how much phase shift the incoming light waves encounter across a particle compared to that in its absence. We systematically investigate variations in $\rho_{BC}$ across the multi-space distribution of BC morphology, mixing-state, mass, and composition as reported by field and laboratory observations. We find that $\rho_{BC} > 1$ leads to decreased absorption by BC, which explains the weaker absorption enhancements observed in certain regional BC compared to laboratory results of similar mixing state. We formulate universal scaling laws centered on $\rho_{BC}$ and provide physics-based insights regarding core-shell approximation overestimating BC light absorption. We conclude by packaging our framework in an open-source Python application to facilitate community-level use in future BC-related research. The package has two main functionalities. The first functionality is for forward problems, where experimentally measured BC mixing state and assumed BC morphology are input, and the aerosol absorption properties are output. The second functionality is for inverse problems, where experimentally measured BC mixing state and absorption are input, and the morphology of BC is returned. Further, if absorption is measured at multiple wavelengths, the package facilitates the estimation of imaginary refractive index of coating materials by combining the forward and inverse procedures. Our framework thus provides a computationally inexpensive source for calculation of absorption by BC, and can be used to constrain light absorption throughout the atmospheric lifetime of BC.



## 1 Introduction

The contribution of aerosols to global radiative forcing remains one of the largest sources of uncertainty in current climate models (Reidmiller et al., 2018). Much of this uncertainty stems from disagreements between the predicted and observed radiative forcing by carbonaceous aerosols (Bond et al., 2013; Gustafsson and Ramanathan, 2016; Boucher et al., 2016). One of the most climatically relevant carbonaceous aerosols is black carbon (BC). Black carbon is widely considered to be a predominate light absorbing atmospheric constituent (Bond et al., 2013; Bond and Bergstrom, 2006). Despite this, light

absorption by BC is still significantly underestimated in current climate models, which stems from incorrect parameterization of BC optical properties (Bond et al., 2013; Gustafsson and Ramanathan, 2016; Boucher et al., 2016). Estimation of BC light absorption is particularly complicated, given that BC is often internally mixed with other species, which manifest as external coatings (China et al., 2013).

External coatings enhance light absorption by BC through a "lensing effect", in which a portion of the incoming light is

scattered by the coating into the BC core, where it is then absorbed (Chakrabarty and Heinson, 2018; Cappa et al., 2012; Peng et al., 2016; Saliba et al., 2016; Liu et al., 2017; Shiraiwa et al., 2010). However, previous studies on light absorption enhancement due to the lensing effect have had various results. Some studies find high absorption enhancement (up to a factor of 2.5), while others find little to no absorption enhancement with increasing coating amount (Cappa et al., 2012; Saliba et al., 2016; Shiraiwa et al., 2010; Liu et al., 2015; Cappa et al., 2019; Zhang et al., 2018; Denjean et al., 2020; Zanatta et al., 2018;

Xie et al., 2019; Cui et al., 2016). The range of absorption enhancement from previous studies is evident in Figure 1. Fierce *et al.* has found that particle-to-particle heterogeneity reconciles a large portion of the observed discrepancies in light absorption enhancement (Fierce et al., 2020). However, even when particle-to-particle heterogeneity is considered, light absorption enhancement is still overestimated, and previous discrepancies cannot be fully resolved. Fierce *et al.* have also shown that representation of the complex morphology of BC further improves estimation of its optical properties, but systematic

understanding of the effect of BC morphology on light absorption enhancement is understudied.

Here, we take a two-pronged approach to develop a simple yet rigorous unified framework for parameterizing the effects of particle size, morphology, and mixing state on BC light absorption. The first approach involves reducing the aforementioned multivariate space to a single parameter that captures causal relationships between BC's physio-chemical properties and corresponding light absorption. Using this parameter, we next develop universal scaling laws for wavelength-dependent BC

light absorption as a function of size, morphology, and mixing state. We validate these laws against observational datasets from eleven field campaigns which investigated global trends in BC absorption, as well as laboratory experiments that investigated light absorption enhancement. From the standpoint of practical applications of our framework, we package our scaling laws into an open-source python software which allows researchers to use our results to estimate absorption of BC aerosols based on their size, morphology, and mixing state, and also estimate the morphology of BC aerosols based on their

size, absorption, and mixing state.





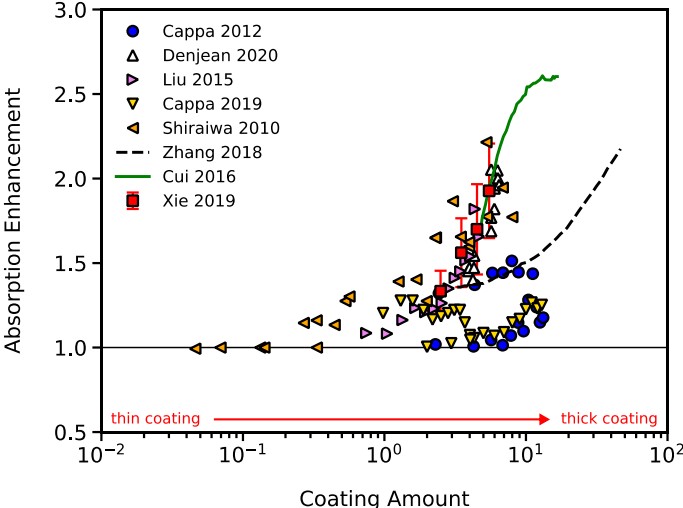

**Figure 1:** Results from previous studies on light absorption enhancement. Some studies find high absorption enhancement, which have had large discrepancies. Some studies find that absorption is enhanced by over twofold, while other studies find little to no absorption enhancement with increased coating amount
(Cappa et al., 2012; Denjean et al., 2020; Liu et al., 2015; Cappa et al., 2019; Shiraiwa et al., 2010; Zhang et al., 2018; Cui et al., 2016; Xie et al., 2019)**.**

## 2 Methods

### 2.1 Representation of diverse black carbon morphologies

Black carbon is often modeled assuming a spherical core-shell configuration. However, soon after emission, BC aggregates have been found to have a lacy, fractal-like structure. Surface tension and capillary forces from the buildup of external coatings
can cause BC aggregates to collapse, and eventually take on a more spherical structure (China et al., 2013; Liu et al., 2017; Fierce et al., 2020; Wang et al., 2017). Recent studies have found that non-sphericity of BC containing particles (partial encapsulation of BC) can decrease absorption enhancement (Hu et al., 2022, 2021). While these findings are notable, previous studies have not observed a prevalence of partially-encapsulated BC, yet decreased light absorption enhancement is still observed (China et al., 2013; Fierce et al., 2020). Therefore, this study is focused on investigating the effects of core
restructuring on light absorption enhancement, rather than the effects of partial BC encapsulation.

To model the evolution of BC morphology, we utilize three aggregation models which represent fresh, partially collapsed, and fully collapsed BC aggregates. Fresh BC aggregates were created using an off-lattice diffusion-limited cluster-cluster aggregation model, which has been shown to accurately represent BC aggregates produced by combustion systems (Meakin, 1983, 1987). Partially and fully collapsed BC aggregates were respectively simulated with a percolation model and simple
cubic lattice stacking. These particles resemble electron microscope images of moderately and heavily coated BC, respectively (Fierce et al., 2020). Each simulated BC particle is comprised of monomers with radius equal to 20 nm (Bond et al., 2013). The amount of coating was quantified by the ratio of coating mass to BC mass ($R_{BC}$). Under this definition, increased $R_{BC}$





represents increasing coating amount, and $R_{BC} = 0$ represents pure BC. The mass of the BC core and the coating material were determined per their volume and densities, 1.8 g/cm$^3$ and 1.2 g/cm$^3$, respectively (Bond and Bergstrom, 2006). This study

utilized 345 aggregates, with BC masses between ~1 fg and ~70 fg, and $R_{BC}$ between 0 and 49. Figure 2 (panels a and c) shows examples of simulated fresh and fully collapsed BC aggregates, which represent the extremes of observed BC morphology. Qualitatively, the morphology of partially collapsed BC aggregates lies between that of fresh and fully collapsed aggregates. An example of a partially collapsed aggregate can be found in supplementary figure S1.

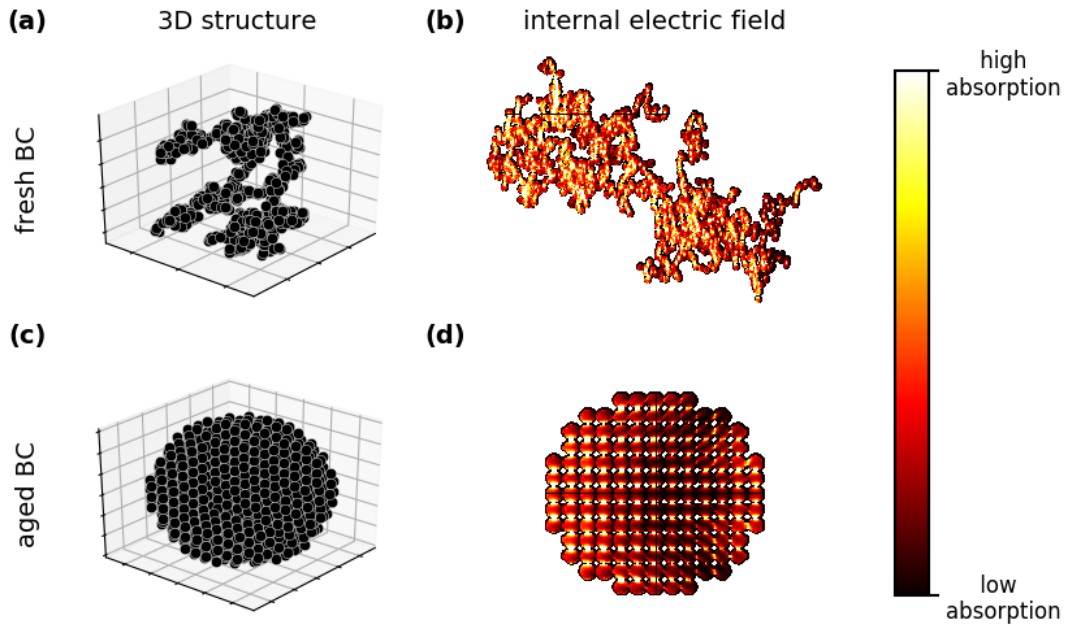

**Figure 2:** The first column shows examples of 3D structure of (a) freshly emitted (open aggregate) and (c) aged (collapsed aggregate) black carbon (BC) particles, which represent the extremes of BC morphology observed as a function of atmospheric processing in field and laboratory studies. The second column shows internal light absorption distribution across these aggregates, with light incident from the left of the particle. As BC becomes more compact, areas of decreased light absorption begin to emerge in the interior of the aggregate, which in turn leads to decreased MAC$_{BC}$.

## 2.2 Calculation of optical properties

The optical properties of the generated aggregates were calculated using the Amsterdam Discrete Dipole Approximation (ADDA 1.3b4) algorithm (Yurkin and Hoekstra, 2011). The ADDA algorithm calculated the absorption cross-section of each aggregate, which was then divided by the mass of the BC core to give mass absorption cross-section (MAC$_{BC}$) of each aggregate. Much of the previous work which investigates light absorption by internally mixed BC measure absorption enhancement ($E_{abs}$). Absorption enhancement is commonly defined as absorption by internally mixed BC divided by absorption

by pure BC (Fierce et al., 2020). There are three common methods for estimating light absorption by pure BC: direct



measurement (using thermodenuders to remove coating material), extrapolation of best fit lines of light absorption by internally mixed BC, and using literature values. All of these methods have challenges which can ultimately affect the reported value of $E_{abs}$. It has been found that thermodenuders may not remove low-volatility coating material, which leads to overestimation of light absorption by pure BC and underestimation of $E_{abs}$ (Shetty et al., 2021). Extrapolation of absorption measurements by internally mixed BC either assume that the morphology of BC does not affect light absorption, or that the morphology of BC remains fixed as coating accumulates. Finally, use of literature values to approximate light absorption by pure BC assumes that light absorption by fractal aggregates is equivalent to literature values of absorption by bulk BC. Rayleigh-Debye-Gans approximation of light absorption by fractal BC is significantly lower than commonly used literature values of absorption by pure BC (Bond and Bergstrom, 2006; Sorensen, 2001), indicating that use of literature values can also underestimate $E_{abs}$.

In order to avoid the errors associated with measurement of $E_{abs}$, we instead focus our efforts on quantification of $MAC_{BC}$. Absorption cross-section per BC mass is a common input of radiative transfer algorithms, and is vital in converting BC mass concentration to absorption coefficient (Bond et al., 2013). Accurate scaling of $MAC_{BC}$ as a function of aggregate size, morphology, and mixing state will allow for subsequent calculation of $E_{abs}$ which accounts for the evolution of BC morphology throughout its atmospheric lifetime.

**2.3 Phase shift parameter is a unifying measure of size, morphology, and composition**

Previous studies of $E_{abs}$ have focused on the effects of a single dependent variable (absorption) as a function of a single independent variable (mixing state). However, detailed representation of the microphysical properties of BC leads to the introduction of several other measures which describe the size and morphology of BC, increasing the size of the variable set from two (absorption and mixing state) to four (size, morphology, absorption, and mixing state).

To reduce the size of the variable set, we utilize the phase shift parameter ($\rho$), which is a unifying measure of both aggregate size and morphology. Physically, $\rho$ describes the amount of phase shift that light accumulates when passing through a particle (Heinson and Chakrabarty, 2016; Sorensen and Fischbach, 2000). When $\rho$ is less than one, there is not a significant amount of phase shift in the incident wave, and the particle-light interactions are well described by Rayleigh approximations. Conversely, when $\rho$ is greater than one, the particle-light interactions are well described by geometric optics (Sorensen and Fischbach, 2000). In this work, $\rho$ is used to describe the size and morphology of the BC core, not the entire particle (BC core + coating). Therefore, in the remaining text we refer to the core phase shift parameter ($\rho_{BC}$) to distinguish from the phase shift parameter of the entire particle. The core phase shift parameter is given by (Debye, 1958)

$$\rho_{BC} = \frac{4\pi R_g}{\lambda} |m_{eff} - 1|, \tag{1}$$



Where $\lambda$ is the wavelength of incident light, $R_g$ is the particle radius of gyration (size metric), and $m_{eff}$ is the effective complex
index of refraction, which in turn is given by

$$\phi\left(\frac{m^2-1}{m^2+2}\right) = \left(\frac{m_{eff}^2-1}{m_{eff}^2+2}\right). \tag{2}$$

Here, $\phi$ is the BC monomer packing fraction and $m$ is the BC complex refractive index. The BC refractive index is fixed at
1.95+0.79i (Bond and Bergstrom, 2006). The BC monomer packing fraction was calculated as the volume of BC which lies
within a sphere of radius $R_g$ (centered at the center of mass) divided by the volume of a sphere with radius $R_g$. It is important
to note that all parameters used in equation 1 and 2 describe the BC core, not the entire particle. It is also important to note
that the dynamic compositional changes which a BC particle undergoes during atmospheric processing are captured by $m_{eff}$ in
equation 2 (Heinson et al., 2017). As BC becomes more internally mixed, manifested as thicker shell coatings, two competing
processes occur (Fierce et al., 2020). The first is that voids within BC aggregates become filled with coating material, leading
to decreased $\phi$. The second is that the BC core will begin to collapse due to surface tension and capillary forces, and $\phi$ will
increase. Both of these changes will affect $m_{eff}$ and eventually $\rho_{BC}$.

The consequence of increased $\rho_{BC}$ on light absorption can be seen in Figure 2(b) and (d), which show internal fields of the BC
aggregates shown in Figure 1(a) and (c). For fresh aggregates (with $\rho_{BC} \ll 1$), light is able to fully illuminate the aggregate,
and the entire volume contributes to light absorption. However, for fully collapsed aggregates (with $\rho_{BC} > 1$), light is not able
to illuminate the far interior of the particle, leading to areas of decreased light absorption. Therefore, if $\rho_{BC}$ of a particle
significantly increases, its light absorption properties will change significantly. It should be noted that since $\rho_{BC}$ is a function
of both morphology and size of aggregates, full core collapse will not always lead to $\rho_{BC} > 1$. Aggregates with a small number
of monomers may never achieve $\rho_{BC} > 1$, even when the monomer packing fraction reaches unity.

## 3 Results and Discussion

### 3.1 Sensitivity of MAC$_{BC}$ to coating refractive index






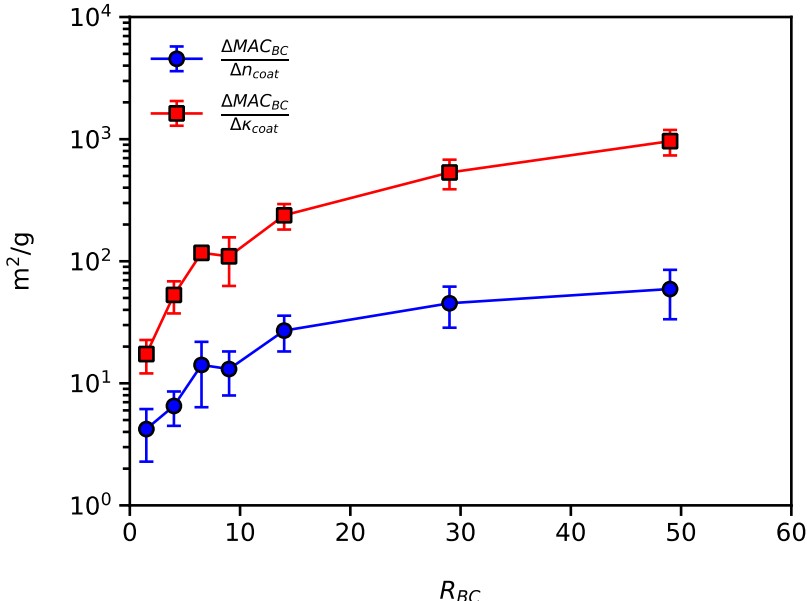

**Figure 3:** Change in MAC$_{BC}$ per change in coating real refractive index ($\Delta$MAC$_{BC}$/$\Delta n_{coat}$) and change in MAC$_{BC}$ per change in coating imaginary refractive index ($\Delta$MAC$_{BC}$/$\Delta\kappa_{coat}$). We find that $\Delta$MAC$_{BC}$/$\Delta\kappa_{coat}$ is from 4 to 16 times greater than $\Delta$MAC$_{BC}$/$\Delta n_{coat}$, depending on R$_{BC}$. These results imply that the choice of coating imaginary refractive index is more important than the choice of coating real refractive index when calculating MAC$_{BC}$.


To examine the effects of coating refractive index, we calculate MAC$_{BC}$ of 30 randomly selected BC aggregates with coating real refractive index ($n_{coat}$) of 1.45, 1.55, and 1.65, and coating imaginary refractive index ($\kappa_{coat}$) of 0.00, 0.05, and 0.1. Figure 3 shows the change in MAC$_{BC}$ per change in $n_{coat}$ ($\Delta$MAC$_{BC}$/$\Delta n_{coat}$) and change in MAC$_{BC}$ per change in $\kappa_{coat}$ ($\Delta$MAC$_{BC}$/$\Delta\kappa_{coat}$). We find that $\Delta$MAC$_{BC}$/$\Delta\kappa_{coat}$ is always greater than $\Delta$MAC$_{BC}$/$\Delta n_{coat}$, and increases with increased R$_{BC}$. These results show that

the choice of $\kappa_{coat}$ is more important than the choice of $n_{coat}$ when calculating MAC$_{BC}$. Given this, we further investigate scaling of MAC$_{BC}$ with $\kappa_{coat}$ between 0.00 and 0.05, but coating $n_{coat}$ remained fixed at 1.55 (Bond and Bergstrom, 2006). In the context of field and laboratory measurements, particles with $\kappa_{coat}$ = 0.00 are representative of BC which is internally mixed with non-refractory material, such as $\alpha$-pinene and sulfuric acid (Fierce et al., 2020). Particles with $\kappa_{coat}$ > 0.00 are representative of BC which is internally mixed with absorbing material, such as brown carbon (Liu et al., 2015).

**3.1 Phase shift parameter controls light absorption**

Figure 4 shows MAC$_{BC}$ as a function of $\rho_{BC}$ and R$_{BC}$ for incident wavelength ($\lambda$) of 532 nm. Panels b, d, and f of Figure 4 show the clear emergence of two regimes separated by $\rho_{BC}$ = 1 (dashed line). For $\rho_{BC} \leq 1$, MAC$_{BC}$ increases with increased R$_{BC}$, but is independent of $\rho_{BC}$. For $\rho_{BC} > 1$, MAC$_{BC}$ decreases with increased $\rho_{BC}$, and the rate of decrease is dependent on R$_{BC}$. The finding of decreased light absorption for $\rho_{BC} > 1$ is consistent with a recent study which also found decreased MAC$_{BC}$

with increasing aggregate size (Romshoo et al., 2021). Best-fit lines for the scaling of MAC$_{BC}$ as a function of R$_{BC}$ are shown as solid lines in Figure 4, and are summarized by

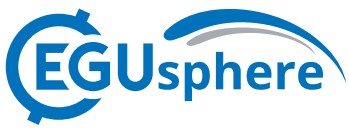

$$\frac{\partial MAC_{BC}}{\partial R_{BC}} = \begin{cases} MAC_0 \left(\frac{\lambda}{\lambda_0}\right)^{-AAE} \left\{\frac{1}{C}[AC^{-B}\boldsymbol{\Gamma}(B+1,C)] - 2 - \frac{1}{C}\left[A(R_{BC}+1)^B \left(C(R_{BC}+1)\right)^{-B}\boldsymbol{\Gamma}\left(B+1,C(R_{BC}+1)\right)\right] + 3\kappa_{coat}(R_{BC}+1)\right\}, \rho_{BC} \leq 1 & (3a) \\ MAC_0 \left(\frac{\lambda}{\lambda_0}\right)^{-AAE} \left\{\frac{D}{E-1}\left[\left(\frac{1}{\rho_{BC}}\right)^{E-1}-1\right] + \frac{D}{1-2E}\left[\left(\frac{1}{\rho_{BC}}\right)^{2E-1}-1\right]\right\} + MAC_{BC}(R_{BC},\rho_{BC} \leq 1), \quad \rho_{BC} > 1 & (3b) \end{cases}$$

Where A, B, and C are constants, $\boldsymbol{\Gamma}$ is the incomplete gamma function, and $\kappa_{coat}$ is the imaginary part of the coating refractive index. The fitting parameters D and E are functions of $R_{BC}$, given by

$$X = x_1 + \frac{x_2 - x_1}{1 + exp[x_3(R_{BC} - x_4)]}. \tag{4}$$

Where X generically represents D or E, and $x_{[1,2,3,4]}$ denotes $d_{[1,2,3,4]}$ or $e_{[1,2,3,4]}$. Finally, $MAC_0$ is the average $MAC_{BC}$ at $\lambda = 532$ nm of uncoated aggregates with $\rho_{BC} \leq 1$ (6.819 m$^2$/g) and AAE is the absorption Angstrom exponent for pure BC (1.158). The value of all constants used in equations 3 and 4 can be found in Table 1. Details regarding the acquisition of AAE can be found in supplementary figure S2.







**Figure 4.** Data and fitting of MAC$_{BC}$ as a function of $\rho_{BC}$ and R$_{BC}$ for BC internally mixed with coating imaginary refractive index ($\kappa_{coat}$) of 0.00 **(a-b)**, 0.01 **(c-d)**, and 0.05 **(e-f)**. We find that for constant $\rho_{BC}$, MAC$_{BC}$ increases with increasing R$_{BC}$. Additionally, for constant R$_{BC}$, MAC$_{BC}$ decreases when $\rho_{BC}$ surpasses unity. Solid lines show scaling of MAC$_{BC}$ given by equation 3.





We find AAE which is consistent with previously reported values (Bond et al., 2013; Romshoo et al., 2021), and fitting parameter B which is consistent with a previous numerical study of coated BC aggregates with $\rho_{BC} \leq 1$ (Chakrabarty and Heinson, 2018). The value of $MAC_0$ is less than commonly used literature values of pure BC (7.75 m²/g) (Bond et al., 2013), but slightly greater than Rayleigh-Debye-Gans approximation for $MAC_{BC}$ of fractal aggregates (5.01 m²/g) (Sorensen, 2001). Figure 5(a) shows residual plots for the fitting of $MAC_{BC}$ using equation 3. On average, equation 3 overestimates $MAC_{BC}$ at $\lambda$ = 532 nm by 0.26%, with standard deviation of 5.45%. Generally, relative errors increase as $R_{BC}$ increases, as shown by Figure 5(b). However, equation 3 does not consistently overestimate or underestimate $MAC_{BC}$.

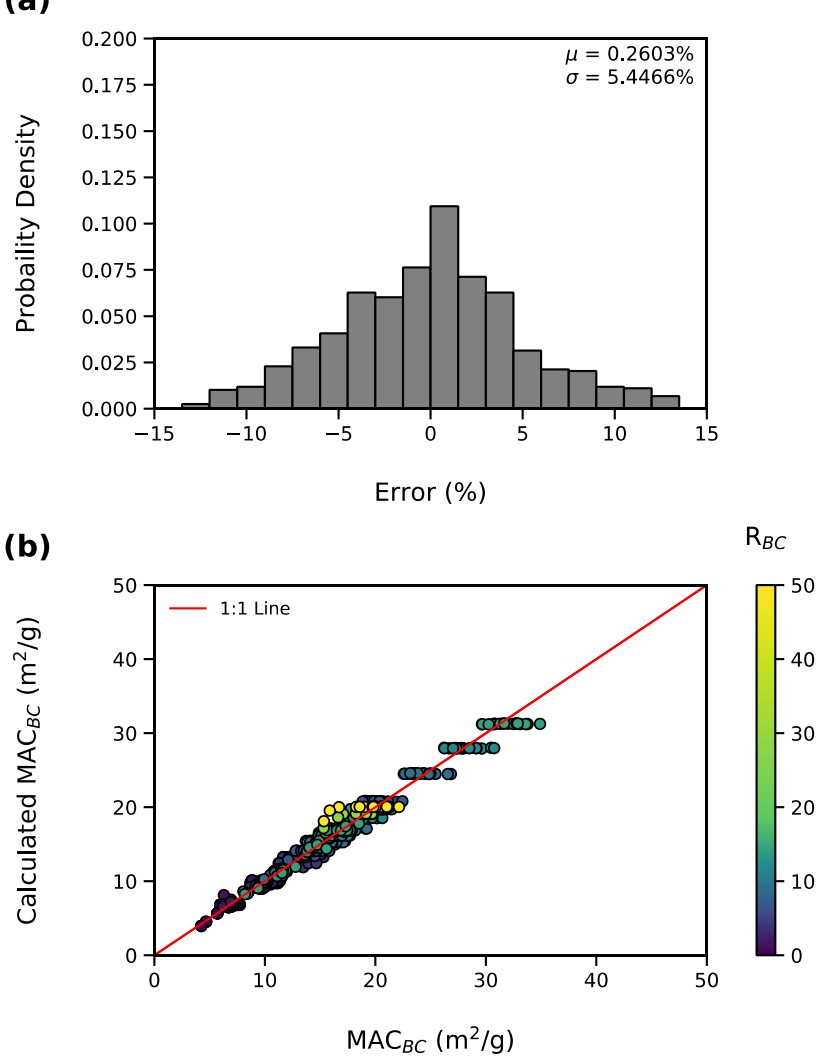

**Figure 5. (a)** Residual plot of fitting of equation 3. On average, equation 3 overestimates $MAC_{BC}$ by 0.26%, with standard deviation of 5.45% (μ and σ, respectively). **(b)** $MAC_{BC}$ calculated using equation 3 as a function of $MAC_{BC}$ from ADDA. Equation 3 accurately predicts $MAC_{BC}$ from the mixing state and morphology of the BC aggregates used to develop equation 3, but relative errors increase with increased $R_{BC}$.




**Table 1.** Value of constants used for fitting of $MAC_{BC}$.

| Constant | Value | 95% CI |
|---|---|---|
| A | -1.189 | 0.029 |
| B | -0.674 | 0.006 |
| C | 0.043 | 0.0007 |
| $d_1$ | 5.679 | 0.027 |
| $d_2$ | 1.066 | 0.058 |
| $d_3$ | 0.264 | 0.010 |
| $d_4$ | 11.421 | 0.137 |
| $e_1$ | 2.440 | 0.017 |
| $e_2$ | 0.593 | 0.024 |
| $e_3$ | 0.418 | 0.020 |
| $e_4$ | 10.106 | 0.131 |
| $MAC_0$ ($m^2/g$) | 6.819 | 0.131 |
| AAE | 1.158 | 0.028 |


### 3.2 Wavelength dependency and limitations of core-shell Mie theory

Coated BC is conventionally modeled with a core-shell morphology, using Mie theory to calculate its light absorption properties (Bond and Bergstrom, 2006). Our results indicate that misrepresentation of BC morphology will inevitably lead to errors in calculation of its light absorbing properties. To highlight this point, Figure 6 compares the accuracy equation 3 in

calculating $MAC_{BC}$ of 15 randomly selected aggregates with $\lambda$ = 405 nm, 532 nm, 880 nm, and 1200 nm to Mie theory calculations for mass-equivalent spheres (with $\kappa_{coat}$ = 0.00). Figure 6 shows that across wavelengths, equation 3 accurately calculates $MAC_{BC}$, with average error of 4.03 ± 10.94%. Conversely, Mie theory is much less reliable in calculating $MAC_{BC}$, with average error of -7.87 ± 22.71%. Most notably, we find that Mie theory consistently overestimates light absorption by BC aggregates with $\rho_{BC}$ > 1, with error reaching over 70% for $\rho_{BC} \approx 3$. This observation is in line with several previous studies

which find that Mie theory greatly overestimates $MAC_{BC}$ of coated BC (Cappa et al., 2019; Fierce et al., 2020).

It should be noted that the development of equation 3 involved only data points for $\lambda$ = 532 nm. However, previous work has shown that enhancement of $MAC_{BC}$ is independent of $\lambda$ (Chakrabarty and Heinson, 2018), indicating that results obtained for $\lambda$ = 532 nm are applicable to other wavelengths. Therefore, Figure 6 (c-d) also shows the utility of equation 3 in calculating $MAC_{BC}$ across wavelengths.







**Figure 6. (a-b)** MAC$_{BC}$ calculated using Mie theory (calculated MAC$_{BC}$) as a function of MAC$_{BC}$, and error incurred by using Mie theory as a function of core phase shift parameter ($\rho_{BC}$). **(c-d)** MAC$_{BC}$ calculated using equation 3 (calculated MAC$_{BC}$) as a function of MAC$_{BC}$, and error incurred by using equation 3 as a function of $\rho_{BC}$. On average, equation 3 predicts MAC$_{BC}$ to within 4.03 ± 10.94%, while Mie theory predicts MAC$_{BC}$ with average error of -7.87 ± 22.71%. Equation 3 is particularly more accurate than Mie theory for aggregates with $\rho_{BC} > 1$, indicating that $\rho_{BC}$ is vital in estimating MAC$_{BC}$ throughout the lifetime of BC.

## 3.3 Validation of scaling laws with field and laboratory observations

Figure 7(a) shows the scaling of MAC$_{BC}$ with R$_{BC}$ for aggregates with $\rho_{BC} \leq 1$, along with data from studies which find significant increases of MAC$_{BC}$ with increasing R$_{BC}$ (Yu et al., 2019; Saliba et al., 2016; Liu et al., 2015; Xie et al., 2019; Denjean et al., 2020; Zanatta et al., 2018). We find that MAC$_{BC}$ from these studies closely matches the behavior of equation 3a, indicating that these studies were measuring light absorption properties of aggregates with $\rho_{BC} \leq 1$. It has been hypothesized that large values of MAC$_{BC}$ in these studies could be the result of absorbing coatings. However, we find that the data from these studies closely matches scaling of MAC$_{BC}$ with $\kappa_{coat}$ fixed at 0.00. The assumption that $\kappa_{coat} = 0.00$ is bolstered by the





fact that refractory organics absorb preferentially at ultraviolet wavelengths (Chakrabarty et al., 2010; Sumlin et al., 2018; Kirchstetter et al., 2004; Sengupta et al., 2018; Shamjad et al., 2018), and we have only included data from visible and near-infrared wavelengths.

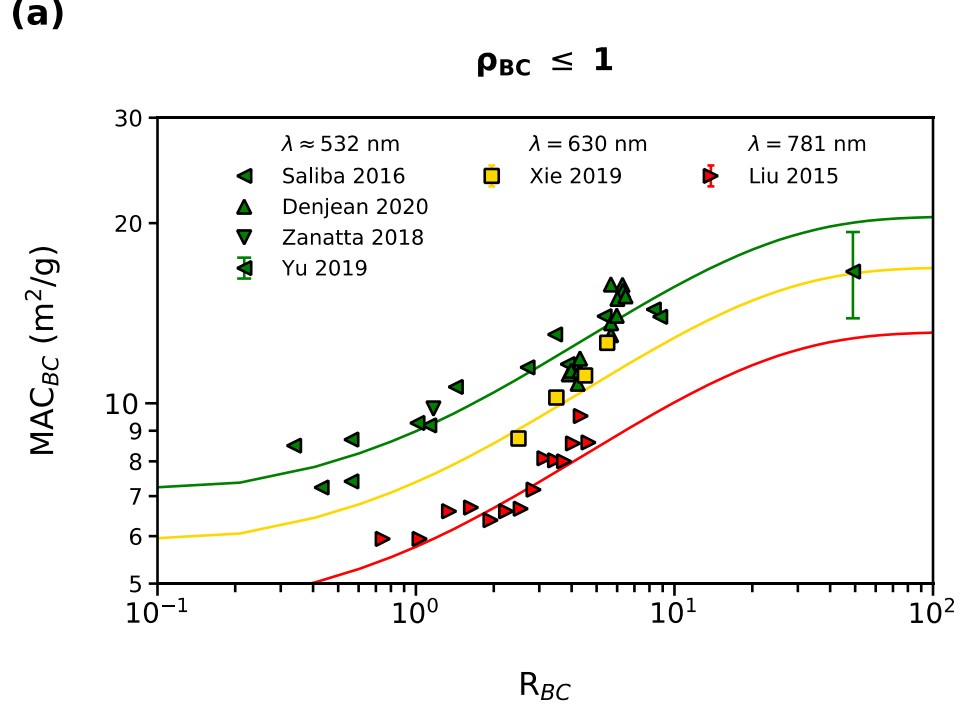

**Figure 7. (a)** Scaling of $MAC_{BC}$ with $R_{BC}$ for aggregates with $\rho_{BC} \leq 1$ (equation 3a, solid lines), compared to the findings of several other studies which find significant $MAC_{BC}$ (Yu et al., 2019; Saliba et al., 2016; Liu et al., 2015; Xie et al., 2019; Denjean et al., 2020; Zanatta et al., 2018). We find that measurements from these studies are well described by equation 3a, indicating that the aggregates measured in these studies had $\rho_{BC} \leq 1$. **(b)** Previous studies which find little to no increase in $MAC_{BC}$ with increasing $R_{BC}$, and the corresponding average $\rho_{BC}$ which replicates the measured $MAC_{BC}$. Using equation 3b, we are able to estimate $\rho_{BC}$ of aggregates from studies which find values of $MAC_{BC}$ which are significantly lower than that predicted by equation 3a (Cappa et al., 2012;
Shiraiwa et al., 2010; Cappa et al., 2019; Zhang et al., 2018; Cui et al., 2016). Error in $\rho_{BC}$ is one standard-deviation.





Figure 7(b) shows a table of previous studies which find little to no increase in $MAC_{BC}$ with increasing $R_{BC}$, and the corresponding average $\rho_{BC}$ which replicates the measured $MAC_{BC}$. The average $\rho_{BC}$ was found by solving equation 3b, inserting each measured $R_{BC}$ and corresponding $MAC_{BC}$. The importance of $\rho_{BC}$ in estimation of $MAC_{BC}$ is evident when
comparing experimentally-measured $MAC_{BC}$ for different studies shown in Figure 3(b). For example, Cappa *et al.* 2019 sampled coated BC aggregates in Fontana and Fresno, California, and find little to no increase in $MAC_{BC}$, even with $R_{BC} > 10$ (Cappa et al., 2019). They postulate that the low value of $MAC_{BC}$ is due to unequal distribution of coating material between BC particles. Separate studies have shown that uneven distribution of coating can cause decreased $MAC_{BC}$, but thorough consideration of heterogeneous coating amounts fails to fully explain low $MAC_{BC}$ observed in the field (Fierce et al., 2020).
Therefore, our results suggest that elevated $\rho_{BC}$ due to core restructuring may be partially responsible for low $MAC_{BC}$ observed by Cappa *et al.* 2019*,* further highlighting the importance of the diversity of BC morphology *and* mixing state in estimation of its light absorption properties.

### 3.4 Applications of the developed framework

The scaling laws given in this work allow experimentalists to carry out two procedures. The first is the forward procedure,
where experimentally-measured BC mass, mixing state, and coating refractive index are combined with assumed BC morphology, and $MAC_{BC}$ is calculated. The second is the inverse procedure, where experimentally-measured BC mass, mixing state, and $MAC_{BC}$ are inputs and BC morphology is output. Further, the inverse and forward procedures can be combined to estimate $\kappa_{coat}$. We have developed an open-source python package, called the 'python BC absorption package' (pyBCabs), which performs the forward and inverse functions. The following sections provide a brief overview of pyBCabs, as well as
examples of inverse problems and estimation of $\kappa_{coat}$. Further details regarding the functionality of the package, as well as more examples of forward and inverse problems for single BC particles and distributions of BC particles can be found at https://pybcabs.readthedocs.io/en/latest/index.html.

### 3.4.1 Forward procedure for light absorption properties

In the forward procedure, experimentally measured $R_{BC}$ and single particle BC mass are first combined with the assumed
morphology of BC and $\kappa_{coat}$. Then, $\rho_{BC}$ is calculated using equations 1 and 2, based on the assumed morphology and BC mass. Finally, $MAC_{BC}$ is calculated using equation 3(a) or 3(b). A flowchart of the forward procedure is shown by the dashed lines in Figure 8(a). As an example, a fresh BC particle with mass-equivalent diameter of 300 nm and $R_{BC} = 3.68$ is input to the forward procedure along with the wavelength of interest (405 nm), and $MAC_{BC} = 16.703$ m$^2$/g is output. A BC particle with the same characteristics of the previous example, but with a fully collapsed BC core would have $MAC_{BC}$ of 11.46 m$^2$/g. This
example demonstrates the utility of the developed framework in evaluating changes in $MAC_{BC}$ as coating-induced restructuring occurs during atmospheric processing.




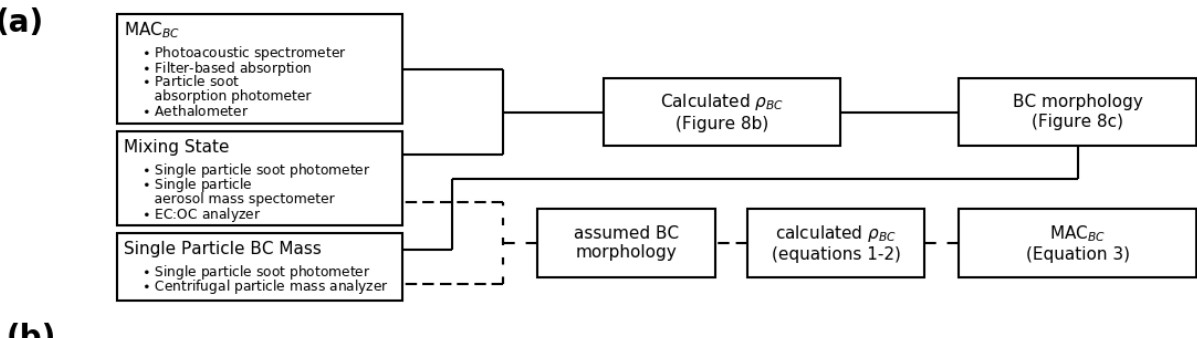

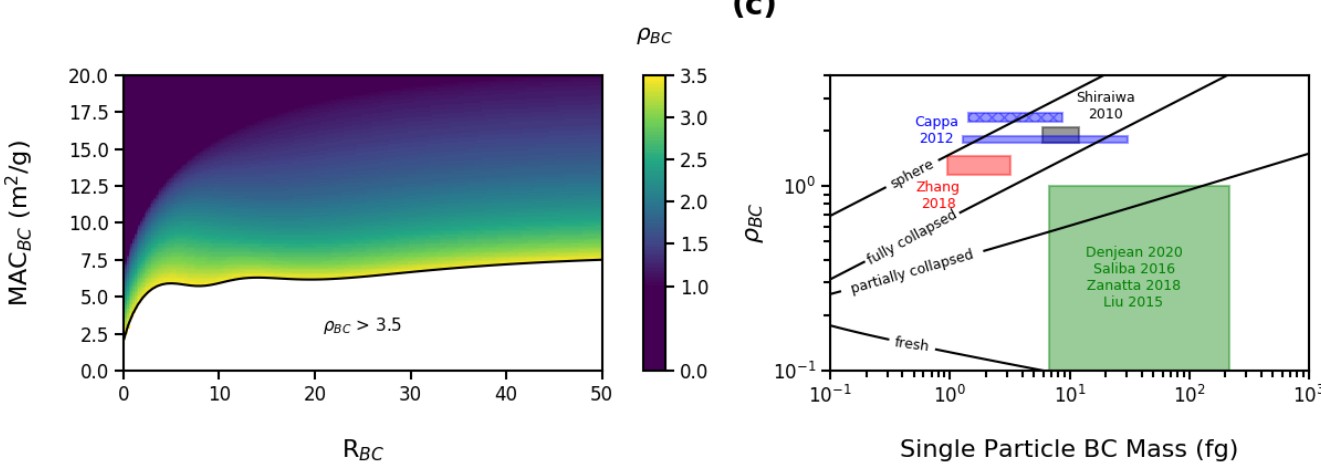

**Figure 8. (a)** Flowchart for forward (dashed lines) and inverse (solid lines) procedures of python BC absorption module (pyBCabs), which uses three measured properties: mass absorption cross-section (MAC$_{BC}$), coating amount (R$_{BC}$), and single particle BC mass. In the forward procedure, the assumed BC morphology is combined with the single particle BC mass to calculate MAC$_{BC}$ (dashed lines). In the inverse procedure (solid lines), measurements of MAC$_{BC}$ and R$_{BC}$ are first input to panel **(b)** in order to constrain the core phase shift parameter ($\rho_{BC}$). Then, $\rho_{BC}$ and the single particle BC mass are input to panel **(c)** to estimate the BC core structure. This procedure has been carried out for data from three studies, and find that low MAC$_{BC}$ from these studies can be explained by extensive compaction of the BC core (Cappa et al., 2012; Shiraiwa et al., 2010; Zhang et al., 2018). The procedure outlined in panel (a) has also been carried out for data from several studies which found significant increases in MAC$_{BC}$ with increasing R$_{BC}$, and find that the core morphology lies between fresh and partially collapsed (Saliba et al., 2016; Liu et al., 2015; Denjean et al., 2020; Zanatta et al., 2018). The examples shown here are all calculated assuming $\kappa_{coat}$ = 0.00.

### 3.4.2 Inverse procedure for morphology retrieval

In the inverse procedure, experimentally measured R$_{BC}$ and $\kappa_{coat}$ is first input to equation 3a, and MAC$_{BC}$ is calculated. If the calculated MAC$_{BC}$ replicates the measured MAC$_{BC}$, then it can be concluded that the measured BC has $\rho_{BC} \leq 1$, but the exact $\rho_{BC}$ cannot be determined. If the measured MAC$_{BC}$ is much less than that predicted by equation 3a, then Figure 8 (b) can be used to estimate $\rho_{BC}$. Alternatively, equation 3b can be used to calculate $\rho_{BC}$ directly. Finally, the single particle BC mass and $\rho_{BC}$ are combined with Figure 8 (c) to give insight to how much restructuring the BC core has undergone. A flowchart of the inverse procedure is shown by the solid lines in Figure 8 (a). The inverse procedure has been carried out for 7 previous studies (Cappa et al., 2012; Saliba et al., 2016; Shiraiwa et al., 2010; Liu et al., 2015; Zhang et al., 2018; Denjean et al., 2020; Zanatta




et al., 2018), and the results are shown in Figure 8 (c). Our results indicate that studies which find little to no increase in $MAC_{BC}$ with increased $R_{BC}$ are measuring BC aggregates which have undergone significant coating-induced restructuring, while studies that find significant increases in $MAC_{BC}$ are measuring aggregates which have undergone little to no restructuring.

### 3.4.3 Inverse procedure for coating refractive index retrieval

The inverse and forward procedures can be combined to estimate $\kappa_{coat}$, if $MAC_{BC}$ is measured at multiple wavelengths. To accomplish this, $\rho_{BC}$ is first found using the inverse procedure outlined above, using $MAC_{BC}$ measured at a near-infrared wavelength (where $\kappa_{coat}$ can be estimated as 0.00). Then, $\rho_{BC}$, $R_{BC}$, and $MAC_{BC}$ can be used to solve for $\kappa_{coat}$ at near-ultraviolet and visible wavelengths. This procedure is outlined in Figure 9 (a), and has been carried out for Liu *et al.*'s 2015 study, which measured absorption enhancement for BC which was internally mixed with absorbing organics (Liu et al., 2015). We estimate that for this study, $\kappa_{coat} = 0.056$ at $\lambda = 405$ nm. Figure 9 (b) shows data collected by Liu *et al.* at $\lambda = 781$ nm (red points) and $\lambda = 405$ nm (blue points). The solid lines show $MAC_{BC}$ calculated using equation 3, inserting the appropriate $\lambda$ and $\kappa_{coat}$. Our estimation of $\kappa_{coat}$ is slightly greater than the reported $\kappa_{coat}$ in Liu *et al.* However, Liu *et al.* approximated $\kappa_{coat}$ using Rayleigh-Debye-Gans approximations, not direct measurement (Liu et al., 2015). Additionally, our estimation of $\kappa_{coat}$ is consistent with previous studies of refractive index of absorbing organics (Chakrabarty et al., 2010; Sumlin et al., 2018; Kirchstetter et al., 2004; Sengupta et al., 2018; Shamjad et al., 2018).

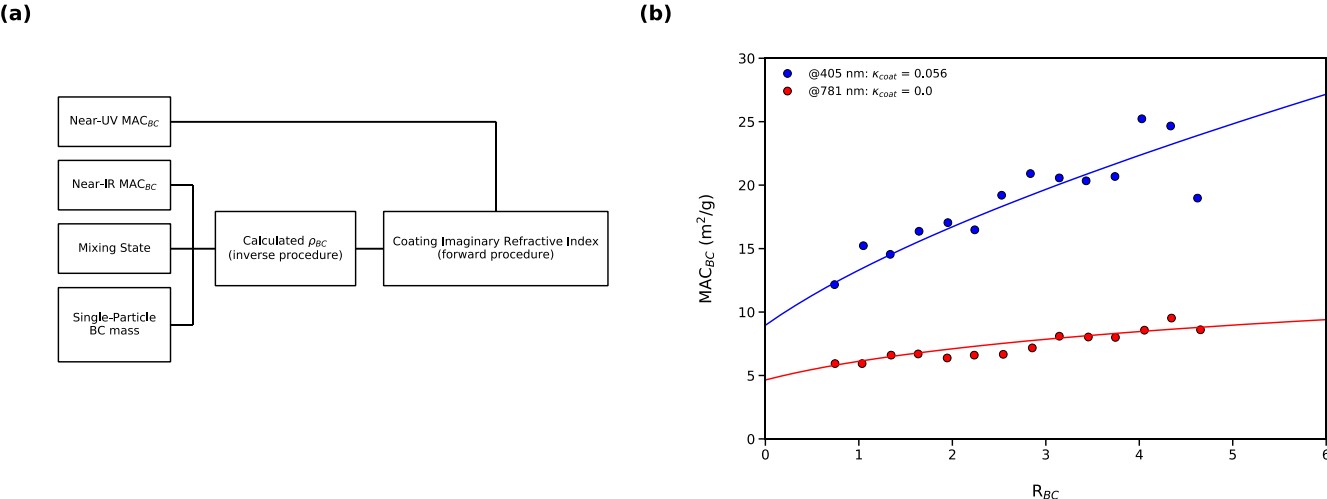

**Figure 9. (a)** Flowchart for estimation of coating refractive index. First, $\rho_{BC}$ is determined using the inverse procedure with $MAC_{BC}$ measured at near-infrared (near-IR) wavelength, where coating imaginary refractive index ($\kappa_{coat}$) can be estimated as 0.00. Then, $MAC_{BC}$ at near near-ultraviolet (near-UV) wavelength and calculated $\rho_{BC}$ are input to equation 3 and $\kappa_{coat}$ is calculated using the forward procedure. **(b)** Example of coating imaginary refractive index retrieval for Liu et al. 2015 (Liu et al., 2015). First, measured $MAC_{BC}$ at $\lambda = 781$ is used to retrieve BC morphology. We find that the BC in this study close matches scaling of $MAC_{BC}$ with $\rho_{BC} \leq 1$. Once $\rho_{BC}$ is known, measured mixing state and $MAC_{BC}$ at $\lambda = 405$ are input to equation 3a and $\kappa_{coat}$ is solved for. We estimate that $\kappa_{coat}$ at $\lambda = 405$ is approximately 0.056. Data points show measurements made by Liu *et al.* and solid lines show result of equation 3 with appropriate $\kappa_{coat}$ and $\lambda$.



## 4 Conclusions

This study comprehensively investigates the effect of BC morphology on light absorption, introduces $\rho_{BC}$ as a central parameter in accurate estimation of MAC$_{BC}$, and develops improved scaling laws for MAC$_{BC}$. We find that for aggregates with $\rho_{BC} \leq 1$,

MAC$_{BC}$ increases with increasing R$_{BC}$. For aggregates with $\rho_{BC} > 1$, MAC$_{BC}$ is a function of R$_{BC}$ and $\rho_{BC}$. Our work also shows that as $\rho_{BC}$ increases past unity, MAC$_{BC}$ decreases. We then provide a comparison of the scaling laws presented in this work with Mie theory calculations for mass-equivalent spheres. We find that Mie theory consistently overestimates MAC$_{BC}$ of internally mixed BC with $\rho_{BC} > 1$, which is consistent with previous studies which also find that Mie theory greatly overestimates absorption by BC (Cappa et al., 2012, 2019; Fierce et al., 2020). The scaling laws presented in this work account

for the microphysical properties of BC, and provide a new tool for estimating BC light absorption based on BC morphology.

Finally, we validate our findings with data from 11 previous studies which measure light absorption enhancement (Yu et al., 2019; Cappa et al., 2012; Saliba et al., 2016; Shiraiwa et al., 2010; Liu et al., 2015; Cappa et al., 2019; Zhang et al., 2018; Xie et al., 2019; Denjean et al., 2020; Zanatta et al., 2018; Cui et al., 2016). We find that studies which find significant absorption

enhancement with increasing R$_{BC}$ agree well with our scaling laws for BC with $\rho_{BC} \leq 1$ (Saliba et al., 2016; Liu et al., 2015; Denjean et al., 2020; Zanatta et al., 2018; Xie et al., 2019; Yu et al., 2019). We also find that $\rho_{BC} > 1$ is a possible explanation for studies which find little to no absorption enhancement (Cappa et al., 2012; Shiraiwa et al., 2010; Cappa et al., 2019; Zhang et al., 2018; Cui et al., 2016). These findings are significant because coating-induced restructuring of the BC core will lead to increases in the core packing fraction, and consequent increases in $\rho_{BC}$. Our findings suggest that restructuring of the BC core

and increased $\rho_{BC}$ can lead to decreased absorption, and may play a role in previous discrepancies in measured MAC$_{BC}$. Previous work has shown that heterogeneity in BC mixing state accounts for a large portion of the discrepancies in measured and modeled BC, but does not fully reconcile previous discrepancies in BC absorption. Our study shows that particle-resolved mixing state and detailed representation of BC morphology are both necessary in order to fully parameterize absorption by internally mixed BC.


In order to make the results of this study readily available to experimentalists, we conclude by providing an open-source Python module, the 'python BC absorption package' (pyBCabs). This package has two functionalities. The first functionality is for forward problems, where BC mass and R$_{BC}$ of ambient and laboratory generated BC are input, and MAC$_{BC}$ is returned. The second functionality is for inverse problems, where BC mass, R$_{BC}$, and MAC$_{BC}$ of ambient and laboratory generated BC are

input, and the morphology of BC is returned. The forward and inverse functionalities can also be combined to estimate the imaginary part of the coating refractive index, if MAC$_{BC}$ is measured at multiple wavelengths.

The inverse functionality of this module allows for in-situ inference of BC morphology, as opposed to ex-situ methods of determining BC morphology, such as electron microscopy. Use of the inverse functionality of pyBCabs will allow for more



detailed studies on the evolution of BC morphology during its atmospheric lifetime. Improved representation of BC morphology, as well as the improved scaling laws developed by this study can then be incorporated into radiative transfer models, and eventually aid in reducing the uncertainty of radiative forcing by carbonaceous aerosols.

**Data availability.** All data from ADDA calculations are available for download at https://github.com/beelerpayton/ADDA_datasets. Full details regarding the functionality of the developed python package can be found at https://pybcabs.readthedocs.io/en/latest/index.html.

**Supplement.** Includes methods for converting absorption enhancement from previous field and laboratory studies to mass absorption cross-section. Also includes two figures showing examples of a modelled partially collapsed BC particle (S1), and data used for calculation of absorption Ångstrom exponent (S2).

**Author contributions.** RKC and PB conceived of the study and its design. RKC provided guidance and supervision for carrying out the research tasks, interpretation of results, and contributed to the preparation of the manuscript. PB performed the data analysis, developed the figures, and led the preparation of the manuscript. PB and RKC were involved in the editing and proofreading of the manuscript.

**Competing interests**. The authors declare that they have no known competing financial interests or personal relationships that could have appeared to influence the work reported in this paper.

## Acknowledgements

This work was supported by the U.S. National Science Foundation (AGS-1926817), the NASA ACCDAM program (NNH20ZDA001N), and the US Department of Energy (DE-SC0021011). Funding for collecting data during the 2010 CARES field campaign in California was provided by the Atmospheric Radiation Measurement (ARM) Program sponsored by the US Department of Energy (DOE), Office of Biological and Environmental Research (OBER). The authors thank Dr. William Heinson for insightful comments and assisting with getting this project off the ground.

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
