# Peer review of "Constraining the particle-scale diversity of black carbon light absorption using a unified framework"

_EGUsphere, 2022_

## Referee Comment (RC2)

This manuscript presented a method to predict the optical properties of internally mixed black carbon. Based on this manuscript, we can use a single factor called the phase shift parameter to estimate the change of light absorption of internally mixed BC due to different morphology, mixing-state, mass, and coating composition. The method also can be used to predict the absorption properties of coating material and morphology of BC core. This study is significant for BC optical research since it can help improve uncertainties in the current climate model related to BC absorption. The manuscript is well written, and the presentation is clear. The topic also fits very well into the scope of the journal. However, I have some comments and questions about the paper. After considering, responding, and revising the manuscript based on them, this paper should be considered for publication.

**Major Comments:**

1. This manuscript discusses the influence of the mixing state of BC and coating on the absorption of the entire particle. Have you conducted any simulation for BC mixed with inorganics (e.g., dust, sea salt) then coated by other materials? How would that affect your model results?
2. For the AADA model, could you provide a short description of it? I suggest adding a table of import parameters you used.
3. The first Section 3.1 is not clear to me.
   a. Does $\Delta MAC_{BC}$ represent the MAC of the entire particle - MAC of BC core? Same question for $\Delta k_{coat}$ and $\Delta n_{coat}$. Please clarify them.
   b. You used 30 BC aggregates. Does that mean for each aggregate, you used a different combination of $n_{coat}$ and $k_{coat}$? Why do you only have 7 points in figure 3? Shouldn't you have 3 points for k and 3 points for n at each $R_{BC}$?
   c. I also noticed that in Fig. 3, at $R_{BC} = 10$, the ratio decreases. Do you have any explanation for that?
   d. I am not surprised to see $\Delta MAC_{BC}/\Delta k_{coat}$ is always greater than $\Delta MAC_{BC}/\Delta n_{coat}$ since k changes over an order of magnitude, while n only changes within 0.2.
   e. You used $k_{coat}$ between 0.00 and 0.05. What is the step size? Moreover, I think the upper limit of k might be too low to represent a highly absorbing coating. I am wondering why you choose 0.05 instead of 0.1? Also, a range of n instead of 1.55 can better represent ambient particles. I recommend adding sensitivity analysis with a broader range of k and n.
4. For equations 3 a and b, how did you come out with this function? Is there any physical meaning? Moreover, does equation 3 only work with k = 0, 0.01, and 0.05?

**Specific comments**

1. L83-85, "The mass of the BC … (Bond and Bergstrom, 2006)." For the density you chose, I suggest using a range instead of single values so that you can represent the wide range of ambient particle density and uncertainties in the literature. Thus, I am also curious to see the dependence or uncertainties related to the density.
2. Figure 2 is very blurry. Please make sure the final version has a higher resolution.
3. Figure 3, What are these error bars represent?
4. In the manuscript, you used ρ to represent the phase shift parameter. However, ρ is usually used for density. I suggest using a different Greek letter.

5. L132-133, "The BC refractive … ,2006)." Could you clarify whether you use this RI value for all wavelengths or just 550 nm?
6. Section number for sub-sections in Sect. 3 need to be corrected.
7. In figure 4, I can see the authors want to keep the same scale for all sub-figures, but there are too many white spaces in b, d, and f, making it very difficult to see the trend in the $\rho<1$ regime. Please consider changing the y-axis limit. Moreover, in b and f, why the number of $R_{BC}$ dots is different from other $R_{BC}$?
8. L232-234, "It has been … fixed at 0.00." Could it be that the coating is absorbing, but $\rho$ is greater than 1?
9. Figure 7 b should be a table, not a figure.
10. L250, should not Figure 3(b) be Figure 7(b)?

---

## Community Comment (CC2)

**Black-carbon phase shift parameter and soot restructuring**

Joel C. Corbin *(Joel.Corbin@nrc-cnrc.gc.ca)*

June 15, 2022

I found Beeler and Chakrabarty (egusphere-2022-163) very interesting, particularly from the perspective of having just finished a review of soot restructuring studies. That review concluded that only solid coatings or coagulation could allow soot to mix internally without restructuring, and is relevant to the interpretation of the results of this manuscript. I will expand this comment here in the conventional review format for clarity.

In this manuscript, B&C apply the phase-shift parameter $\rho$ to the BC core of internal mixtures. $\rho$ is defined relative to the radius of gyration $R_g$, wavelength $\lambda$, and BC monomer packing fraction $\phi$ as

$$\rho = \frac{4\pi R_g}{\lambda}|m_{\text{eff}} - 1| \tag{1}$$

$$\rho = 2x|m_{\text{eff}} - 1| \tag{2}$$

Where $x = 2\pi R_g/\lambda$ is the size parameter and $m_{\text{eff}}$ is

$$\phi\left(\frac{m^2 - 1}{m^2 + 2}\right) = \frac{m_{\text{eff}}^2 - 1}{m_{\text{eff}}^2 + 2} \tag{3}$$

These equations illustrate that $\rho$ is primarily a function of packing fraction $\phi$ and size $R_g$ (or $x$). Packing fraction is a morphological parameter, and $m$ is only expected to be a function of $R_g$ for aggregates smaller than those considered here (Corbin et al., 2021). So, the main concept in this manuscript is come down to the BC-core packing fraction $\phi$.

Based on the above concept, the authors present an excellent discussion of the relationship between MAC, $\rho$, and the ratio of coating-to-BC-mass, $R_{BC}$, for model soot particles. Then, the authors place their work in the context of the literature by considering whether previous measurements of the relationship between MAC and $R_{BC}$ can be attributed to $\phi$. I have two major questions for the authors:

1. Could the authors add a more quantitative discussion of $\phi$? As stated, the entire discussion of the BC-core $\rho$ comes down to $\phi$, which can be constrained as about 0.1 to 0.4 (Zangmeister et al., 2018, http://www.pnas.org/cgi/doi/10.1073/pnas.1403768111; also Schnitzler et al. 2017 is relevant http://dx.doi.org/10.1016/j.jaerosci.2017.01.005). If there was some reason the authors did not discuss $\phi$ directly could they please comment? If not,

   (a) can the authors calculate $\phi$ for their model aggregates, and discuss whether the upper-limit packing density identified by Zangmeister et al. 2018 allows the literature trends to be fully explained by $\phi$?

   (b) Also, can the authors provide more information about the $\phi$ of their modelled particles, for example by plotting $\phi$ versus $\rho$ or MAC?

2. The authors state that "Our results indicate that studies which find little to no increase in $MAC_{\text{BC}}$ with increased $R_{BC}$ are measuring BC aggregates which have undergone significant coating-induced restructuring, while studies that find significant increases in $MAC_{\text{BC}}$ are measuring aggregates which have undergone little to no restructuring." How confident are the authors that alternatives have been excluded, and that this statement is the most likely given all available evidence? For example:

(a) Based on a recent review of soot-restructuring studies (Corbin, Modini, and Gysel-Beer, arXiv.2206.03646) I believe this statement should be reconsidered or discussed in terms of the fundamental physics it implies. To briefly summarize that review, we identified multiple studies that demonstrated unequivocally that liquid condensation typically induces restructuring. These studies used various materials including organics of varying polarity and sulfuric acid. Our review of these studies and complementary laboratory demonstration, showed that condensation-compaction coatings can only be avoided when solid coatings or liquids with very high contact angles (which may result in heterogeneous nanodroplet-activation and therefore avoid compacting surface tension forces) were used. Examples of such solids include SOA formed at low RH or anthracene (relevant only to the laboratory). Compaction can also be avoided by coagulation. So, if internally mixed BC has not undergone extensive restructuring, it must have mixed with solids (including highly viscous glasses) or by coagulation.

My impression from the recent studies by Fierce et al. (cited by the authors) is that while night-time coagulation can be significant, it is unlikely that most soot particles mix by coagulation. But, perhaps the authors' work implies that my impression was inaccurate. My impression is also that solid organic coatings form only rarely, since they require very low RH or low temperatures, while organic vapours are emitted most often at higher temperatures. It seems to me less likely that solid coatings explain the field data on absorption enhancement.

So, the authors' conclusions can be reconciled with the known mechanisms of soot restructuring by arguing that some studies primarily observe liquid-condensation coatings while other primarily observe solid-like coatings or coagulation coatings.

(b) On Line 71, the authors state, "Recent studies have found that the non-sphericity of BC-containing particles (partial encapsulation of BC) can decrease absorption enhancement (Hu et al., 2022, 2021). While these findings are notable, previous studies have not observed a prevalence of partially-encapsulated BC, yet decreased light absorption is still observed". Is it possible that the authors have too readily rejected the hypothesis of H1) partially encapsulated, or off-centre mixing states, in favour of H2) condensation-without-compaction? Given the abovementioned review, I believe H1 is plausible while H2 is extremely unlikely. I would consider the entire manuscript to remain valid and valuable if H1 is rejected over H2. The only change is that $\phi$ becomes $\phi_{\mathrm{eff}}$. (Would the same trends in MAC be observed?)

(c) This is more of an editorial comment. The highest $\rho_{BC}$ in Figure 7b were measured at the shortest wavelengths and the two highest studies were both first-authored by Cappa. Some readers may wonder whether there was a systematic effect here (for wavelength, it is expected by definition; and for the Cappa group, the question is whether they use a unique experimental approach that caused a bias relative to other data sets). I do not personally believe that these are real issues but they deserve may a brief comment for the reader's benefit.

While reading, I also made various minor notes. I will list them below as suggestions for the authors.

(a) I'd add a row showing partially encapsulated/collapsed examples in Figure 2.

(b) Line 105-109 may need clarifying. Why would someone use the RDG MAC when estimating $E_{abs}$? To me, a "literature value" would be a measured MAC of mature, open-structured BC (Liu et al., linked below). Text may not convey your intention here.

(c) What is the role of $\phi$ in Figure 3? No effect?

(d) Line 161, there may be a better citation for the imaginary refractive index (Sun and Bond?).

(e) Line 163, add SOA after pinene.

(f) Line 164, consider citing Lu et al http://dx.doi.org/10.1021/acs.est.5b00211

(g) Line 180, are the four digits of precision meaningful in 6.819 m2/g? What is the corresponding standard deviation? Also, it may be worth discussing this value in comparison to the measured mean value of 8.0 ± 0.7 m2/g (Liu et al. https://doi.org/10.1080/02786826.2019.1676878)

(h) Figure 4 uses both "$\rho$" and "Core Phase Shift Parameter" for the same thing, which confused me initially. Consider harmonizing.

(i) Figure 7a why are there 3 lines? Please label?

(j) Figure 7b consider adding a column of the range of observed $R_{\mathrm{BC}}$?

(k) Line 272 how could a 'fresh BC' particle have $R_{BC} = 3.68$? It seems that 'fresh' is ambiguous. Maybe 'uncompacted'.

(l) Figure 8a consider omitting the instrument lists, which are incomplete and may become outdated in a shorter time than this work will. If you keep it, please revise (e.g. Single particle BC mass can be measured by SP-AMS and all of the "MAC" instruments measure absorption, not MAC.)

(m) Figure 8b consider contours, I could not see the contrast on my B&W printout.

(n) Caption of Figure 8 states that the low MAC of Cappa 2012 can be explained by compaction of the BC core, but Cappa 2012 shows lab data (their Fig 3) where compaction was absolutely expected yet absorption enhancement was still observed. (My expectation is based on the review of restructuring mentioned above, which includes a repeat of their same experiments and cites Ghazi and Olfert who also repeated those experiments.)

(o) Figure 9a I found confusing but the caption I found clear. Consider linearizing the figure.

(p) Discussion at end of 3.4.2 may have to change to reflect the restructuring comments above.

(q) What are the uncertainties in k= 0.056 in Section 3.4.3? Does the code include an uncertainty estimation feature? Monte Carlo? This would be helpful as a way to let users know when they have obtained meaningful results.

(r) Why would coating filling the voids in the BC aggregate change $\phi$, the BC monomer packing fraction? That is only if the BC core collapses, as stated subsequently.

---

## Author Comment (AC2)

This manuscript presented a method to predict the optical properties of internally mixed black carbon. Based on this manuscript, we can use a single factor called the phase shift parameter to estimate the change of light absorption of internally mixed BC due to different morphology, mixing-state, mass, and coating composition. The method also can be used to predict the absorption properties of coating material and morphology of BC core. This study is significant for BC optical research since it can help improve uncertainties in the current climate model related to BC absorption. The manuscript is well written, and the presentation is clear. The topic also fits very well into the scope of the journal. However, I have some comments and questions about the paper. After considering, responding, and revising the manuscript based on them, this paper should be considered for publication.

**Major Comments:**

1) This manuscript discusses the influence of the mixing state of BC and coating on the absorption of the entire particle. Have you conducted any simulation for BC mixed with inorganics (e.g., dust, sea salt) then coated by other materials? How would that affect your model results?

   We have not considered BC mixed with inorganics. This would alter the effective refractive index of the core, and subsequently alter the phase shift parameter and the absorption cross-section. The model developed in this work will not be accurate for aggregates whose properties vary significantly from those outlined in section 2.2. However, the properties given in section 2.2 are commonly used in large scale models.

2) For the ADDA model, could you provide a short description of it? I suggest adding a table of import parameters you used.

   The most important input parameter for ADDA calculations is the number of dipoles per wavelength of incident light. We have added a description of this input to section 2.2 of the revised manuscript.

3) The first Section 3.1 is not clear to me.
   a) Does $\Delta MAC_{BC}$ represent the MAC of the entire particle - MAC of BC core? Same question for $\Delta k_{coat}$ and $\Delta n_{coat}$. Please clarify them.

      $\Delta MAC_{BC} / \Delta \kappa_{coat}$ and $\Delta MAC_{BC} / \Delta n_{coat}$ represent partial derivatives of of $MAC_{BC}$ with respect to the refractive index. We have changed the notation to $\partial MAC_{BC} / \partial \kappa_{coat}$ and $\partial MAC_{BC} / \partial n_{coat}$, and clarified in the text.

   b) You used 30 BC aggregates. Does that mean for each aggregate, you used a different combination of ncoat and kcoat? Why do you only have 7 points in figure 3? Shouldn't you have 3 points for k and 3 points for n at each $R_{BC}$?

      For each aggregate, we calculated the optical properties using 9 combinations of $n_{coat}$ and $\kappa_{coat}$ at 7 different values of $R_{BC}$. The 7 points in figure 3 are representative of the 7 values of $R_{BC}$, and the points and error bars are the average and standard deviation of the

partial derivative of total particle absorption normalized by BC mass with respect to $n_{coat}$ and $\kappa_{coat}$.

c) I also noticed that in Fig. 3, at $R_{BC} = 10$, the ratio decreases. Do you have any explanation for that?

We do not have any insights to the reason for this. However, since these aggregates are randomly selected this may be washed away if more aggregates are used in the averaging.

d) I am not surprised to see $\Delta MAC_{BC}/\Delta k_{coat}$ is always greater than $\Delta MAC_{BC}/\Delta n_{coat}$ since k changes over an order of magnitude, while n only changes within 0.2.

We wanted to show that changes in $\kappa_{coat}$ that have been observed in ambient particles cause large changes in $MAC_{BC}$. However, the real part of the refractive index does not show as much variability. Therefore, we opted to constrain the sensitivity analysis to observations.

e) You used $k_{coat}$ between 0.00 and 0.05. What is the step size? Moreover, I think the upper limit of k might be too low to represent a highly absorbing coating. I am wondering why you choose 0.05 instead of 0.1? Also, a range of n instead of 1.55 can better represent ambient particles. I recommend adding sensitivity analysis with a broader range of k and n.

In figure 3, the step size for $\kappa_{coat}$ is 0.05. This sensitivity analysis is not intended to give robust estimations of $\Delta MAC_{BC}/\Delta\kappa_{coat}$, but to demonstrate that MAC is more sensitive to $\kappa_{coat}$ than to $n_{coat}$. We chose to limit our study to weakly absorbing coatings because highly absorbing coatings will cause accumulation of phase shift as light passes through the coating. It is not clear whether the scaling laws provide in this work will be accurate for highly absorbing coatings, but due to computation limits we have left this for future work.

4) For equations 3 a and b, how did you come out with this function? Is there any physical meaning? Moreover, does equation 3 only work with k = 0, 0.01, and 0.05?

Equations 3a and 3b come from damped power law distributions of $MAC_{BC}$ as a function of $R_{BC}$ and $\rho_{BC}$. Other than $\rho_{BC} = 1$ representing the crossover from Rayleigh to geometric optics, equation 3 is an empirical fit based on previous work by Chakrabarty and Heinson 2018. Equation 3 is accurate for $\kappa_{coat}$ between 0.00 and 0.05, but has not been tested for $\kappa_{coat}$ outside of this range.

**Specific comments**

1. L83-85, "The mass of the BC ... (Bond and Bergstrom, 2006)." For the density you chose, I suggest using a range instead of single values so that you can represent the wide range of ambient particle density and uncertainties in the literature. Thus, I am also curious to see the dependence or uncertainties related to the density.

Uncertainty related to the density can be calculated by multiplying the calculated MAC$_{BC}$ by the ratio of 1.8 g/cm$^3$ to the density in question. We have done this calculation and added a supplementary figure showing MAC$_{BC}$ with densities between 1.6 and 2.0 g/cm$^3$. We have left the main figure text showing BC with density of 1.8 g/cm$^3$, as this is a commonly used value.

2. Figure 2 is very blurry. Please make sure the final version has a higher resolution.

   This will be corrected upon revision.

3. Figure 3, What are these error bars represent?

   They represent on standard deviation. This has been added to the caption.

4. In the manuscript, you used $\rho$ to represent the phase shift parameter. However, $\rho$ is usually used for density. I suggest using a different Greek letter.

   We chose to stay with notation which has been used in previous publications (Sorenson 2011, https://doi.org/10.1080/02786820117868). While we understand that $\rho$ is also used for density, we are hopeful that section 2.3 clarifies the notation.

5. L132-133, "The BC refractive ... ,2006)." Could you clarify whether you use this RI value for all wavelengths or just 550 nm?

   This has been clarified in the revised text.

6. Section number for sub-sections in Sect. 3 need to be corrected.

   This has been corrected.

7. In figure 4, I can see the authors want to keep the same scale for all sub-figures, but there are too many white spaces in b, d, and f, making it very difficult to see the trend in the $\rho<1$ regime. Please consider changing the y-axis limit. Moreover, in b and f, why the number of RBC dots is different from other RBC?

   There were some issues in the orientation averaging of the optical properties of these particles, so they were excluded from figure 4. We have filled in data points where possible.

8. L232-234, "It has been ... fixed at 0.00." Could it be that the coating is absorbing, but $\rho$ is greater than 1?

   This is possible, but unlikely given that these studies all use wavelengths > 532 nm, where there would be very weak absorption by organic coatings.

9. Figure 7 b should be a table, not a figure.

We have moved figure 7b to a table.

10. L250, should not Figure 3(b) be Figure 7(b)?

Yes, this has been corrected.

---

## Author Comment (AC3)

1) Could the authors add a more quantitative discussion of $\phi$? As stated, the entire discussion of the BC-core $\rho$ comes down to $\phi$, which can be constrained as about 0.1 to 0.4 (Zangmeister et al., 2018, http://www.pnas.org/cgi/doi/10.1073/pnas.1403768111; also Schnitzler et al. 2017 is relevant http://dx.doi.org/10.1016/j.jaerosci.2017.01.005). If there was some reason the authors did not discuss $\phi$ directly, could they please comment? If not,

   a) can the authors calculate $\phi$ for their model aggregates, and discuss whether the upper-limit packing density identified by Zangmeister et al. 2018 allows the literature trends to be fully explained by $\phi$?

   The $\phi$ of our model aggregates ranged from 0.026 to 0.52. Full explanation of some literature trends requires packing densities that are larger than the upper limit observed by Zangmeister et al. in their 2017 paper. While it is not impossible for monomers to arrange in a way that achieves these higher packing fractions (face-centered cubic arrangement with $\phi = 0.74$, for example). However, based on recent findings we think it is unlikely that BC monomer would compact to this degree. Therefore, we think that it is extremely likely that other mechanisms are also playing a role in the decreased MAC observed by some studies, most notably heterogeneity in $R_{BC}$. The intention of this work and subsequent discussion is to demonstrate that BC morphology affects light absorption properties, and to provide a tool which eliminates reliance on models which make assumption about BC morphology. We will add further discussion on the packing fraction of our modeled aggregates and the limits of ambient BC packing fraction upon editing.

   b) Also, can the authors provide more information about the $\phi$ of their modelled particles, for example by plotting $\phi$ versus $\rho$ or MAC?

   Since the phase shift parameter is dependent on the size parameter and $\phi$, we will include the following supplemental figure, which shows the size parameter normalized phase shift parameter as a function of $\phi$. This plot also shows the range of $\phi$ which was included in this study.

[Figure]

2) The authors state that "Our results indicate that studies which find little to no increase in MACBC with increased RBC are measuring BC aggregates which have undergone

significant coating-induced restructuring, while studies that find significant increases in MACBC are measuring aggregates which have undergone little to no restructuring." How confident are the authors that alternatives have been excluded, and that this statement is the most likely given all available evidence? For example:

a) Based on a recent review of soot-restructuring studies (Corbin, Modini, and Gysel-Beer, arXiv.2206.03646) I believe this statement should be reconsidered or discussed in terms of the fundamental physics it implies. To briefly summarize that review, we identified multiple studies that demonstrated unequivocally that liquid condensation typically induces restructuring. These studies used various materials including organics of varying polarity and sulfuric acid. Our review of these studies and complementary laboratory demonstration, showed that condensation-compaction coatings can only be avoided when solid coatings or liquids with very high contact angles (which may result in heterogeneous nanodroplet activation and therefore avoid compacting surface tension forces) were used. Examples of such solids include SOA formed at low RH or anthracene (relevant only to the laboratory). Compaction can also be avoided by coagulation. So, if internally mixed BC has not undergone extensive restructuring, it must have mixed with solids (including highly viscous glasses) or by coagulation.

My impression from the recent studies by Fierce et al. (cited by the authors) is that while night-time coagulation can be significant, it is unlikely that most soot particles mix by coagulation. But, perhaps the authors' work implies that my impression was inaccurate. My impression is also that solid organic coatings form only rarely, since they require very low RH or low temperatures, while organic vapours are emitted most often at higher temperatures. It seems to me less likely that solid coatings explain the field data on absorption enhancement.

So, the authors' conclusions can be reconciled with the known mechanisms of soot restructuring by arguing that some studies primarily observe liquid-condensation coatings while other primarily observe solid-like coatings or coagulation coatings.

We agree with all points given above, and that our previous statement needs to be altered. The original intention of our statement was not to imply that some studies are measuring soot which has not restructured, but that some studies are measuring soot which has achieved $\rho_{BC} > 1$ and others are measuring soot with $\rho_{BC} < 1$. We agree with the conclusions of the above comment that it is unlikely for solid coatings to form in ambient conditions, and therefore unlikely that BC morphology would remain unaltered during coating uptake. We believe that studies which find significant absorption enhancement could be measuring particles which have undergone restructuring, but $\rho_{BC}$ has not increased past unity. This could happen when the product of size parameter and packing fraction is small. For example, small soot particles which undergo significant restructuring (large $\phi$) may still have $\rho_{BC} < 1$, and based on our results would show significant absorption enhancement with increasing coating amounts. We will rephrase our previous statement as follows:

"Our results indicate that studies which find little to no increase in $MAC_{BC}$ with increased $R_{BC}$ may be measuring BC aggregates which have undergone significant

coating-induced restructuring, leading to $\rho_{BC} > 1$. On the other hand, studies that find significant increases in $MAC_{BC}$ may be measuring aggregates which have $\rho_{BC} < 1$, and may also be measuring particles which have significant heterogeneity in $R_{BC}$. This does not imply that these studies are measuring BC which has not restructured, only that the product of the size parameter and core packing fraction of BC is not large enough such that $\rho_{BC} < 1$."

b) On Line 71, the authors state, "Recent studies have found that the non-sphericity of BC containing particles (partial encapsulation of BC) can decrease absorption enhancement (Hu et al., 2022, 2021). While these findings are notable, previous studies have not observed a prevalence of partially-encapsulated BC, yet decreased light absorption is still observed". Is it possible that the authors have too readily rejected the hypothesis of H1) partially encapsulated, or off-centre mixing states, in favour of H2) condensation-without-compaction? Given the abovementioned review, I believe H1 is plausible while H2 is extremely unlikely. I would consider the entire manuscript to remain valid and valuable if H1 is rejected over H2. The only change is that $\phi$ becomes $\phi_{eff}$. (Would the same trends in MAC be observed?)

We do not consider here the effects of off-center mixing states, as it is outside the scope of this work. We agree that we should not reject H1 altogether, but the effects of off-center coatings will be more prevalent at small $R_{BC}$, and we do not believe that H1 can fully explain observations of low $E_{abs}$ at large $R_{BC}$. Since our definition of $\phi$ only includes the BC monomers, the $\phi$ of modeled aggregates will be unaffected by coating location for modeled aggregates.

3. This is more of an editorial comment. The highest $\rho_{BC}$ in Figure 7b were measured at the shortest wavelengths and the two highest studies were both first-authored by Cappa. Some readers may wonder whether there was a systematic effect here (for wavelength, it is expected by definition; and for the Cappa group, the question is whether they use a unique experimental approach that caused a bias relative to other data sets). I do not personally believe that these are real issues, but they deserve may a brief comment for the reader's benefit.

We believe that it is possible that the estimated $\rho_{BC}$ for the Cappa studies may be slightly higher than the true $\rho_{BC}$ of the measured particles. This is because the Cappa studies both made use of thermodenuders. Thermodenuders may not completely remove low-volatility organics from BC, causing overestimation of absorption by BC, and subsequently lower $E_{abs}$ (Shetty et al., 2021 https://doi.org/10.1080/02786826.2021.1873909). It is unclear whether low-volatility organics in fact did bias the absorption measurements in these studies, but is a possible explanation for the elevated $\rho_{BC}$.

While reading, I also made various minor notes. I will list them below as suggestions for the authors.

    a) I'd add a row showing partially encapsulated/collapsed examples in Figure 2.
       We will move the partially collapsed BC example shown in figure S1 to the main text.

    b) Line 105-109 may need clarifying. Why would someone use the RDG MAC when estimating Eabs? To me, a "literature value" would be a measured MAC of mature, open-structured BC (Liu et al., linked below). Text may not convey your intention here.
       This section will be altered to reflect the intention that consideration of fractal morphology can affect the MAC of pure BC.

    c) What is the role of $\phi$ in Figure 3? No effect?
       The change in MAC for changing coating refractive index was calculated with constant $\phi$, so there is no effect. This will be clarified in the text.

    d) Line 161, there may be a better citation for the imaginary refractive index (Sun and Bond?).
       This study will be cited upon revision.

    e) Line 163, add SOA after pinene.
       This will be incorporated upon revision.

    f) Line 164, consider citing Lu et al http://dx.doi.org/10.1021/acs.est.5b00211
       This will be incorporated upon revision.

    g) Line 180, are the four digits of precision meaningful in 6.819 m2/g? What is the corresponding standard deviation? Also, it may be worth discussing this value in comparison to the measured mean value of $8.0 \pm 0.7$ m2/g (Liu et al. https://doi.org/10.1080/02786826.2019.1676878)
       The number of significant digits in the calculated $MAC_0$ is limited by the density of BC, therefore it will be changed to 6.8 $m^2$/g upon revision. We will also discuss this finding in comparison with Liu et al.